# Implementation of mass drug administration for neglected tropical diseases in Guinea during the COVID-19 pandemic

**Fatoumata Sakho**[1], **Christelly Flore Badila**[2], **Benoit Dembele**[3]*, **Aissatou Diaby**[1], **Abdoul Karim Camara**[2], **Lamine Lamah**[2], **Steven D. Reid**[4], **Angel Weng**[4], **Brian B. Fuller**[4], **Katherine A. Sanchez**[5], **Achille Kabore**[5], **Yaobi Zhang**[3], **Angela Weaver**[4]

**1** National Neglected Tropical Disease Control Program, Ministry of Health, Conakry, Guinea, **2** Guinea Office, Helen Keller International, Conakry, Guinea, **3** Regional office for Africa, Helen Keller International, Dakar, Senegal, **4** Headquarters, Helen Keller International, New York, New York, United States of America, **5** Family Health International 360, Washington DC, Maryland, United States of America

* bdembele@hki.org

## Abstract

### Background

Guinea reported its first case of COVID-19 on March 12, 2020. Soon thereafter, a national state of emergency was declared, all land borders were closed, schools were shut down, and public gatherings were limited. Many health activities, including field-based activities targeting neglected tropical diseases (NTDs), were paused. The World Health Organization (WHO) issued updated guidance on the resumption of NTD field-based activities on July 27, 2020. In response, the Guinea Ministry of Health (MoH) and its partners planned and resumed mass drug administration (MDA) in mid-August to September 2020 in 19 health districts.

### Methodology/principal findings

A risk-benefit assessment was conducted to identify potential risks associated with the MDA in the COVID-19 context. Following this assessment, a risk mitigation plan with barrier measures was developed to guide MDA implementation. These measures included COVID-19 testing for all national staff leaving Conakry, mask wearing, social distancing of two meters, and hand washing/sanitizing. A checklist was developed and used to monitor compliance to risk mitigation measures. Data on adherence to risk mitigation measures were collected electronically during the MDA. A total of 120 checklists, representing 120 community drug distributor (CDD) teams (two CDDs per team) and 120 households, were completed. Results indicated that washing or disinfecting hands was practiced by 68.3% of CDD teams, compared to 45.0% among households. Face masks to cover the mouth and nose were worn by 79.2% of CDD teams, while this was low among households (23.3%). In 87.5% of households, participants did not touch the dose pole and in 88.3% of CDD teams, CDDs did not touch the hands of the participants while giving the drugs. A large majority of CDD teams (94.2%) and household members (94.2%) were willing to participate in the MDA despite the

**Data Availability Statement:** All mass drug administration coverage data are contained in this paper, and checklist data are available in S1 Database

**Funding:** This work was made possible by the generous support of the American people through the United States Agency for International Development (USAID). The funding was granted to Helen Keller International under the Act to End Neglected Tropical Diseases | West Program, led by FHI 360 in partnership with Helen Keller International, Health and Development International, Deloitte, World Vision, Americares, and the AIM Initiative (a program of American Leprosy Missions) under Cooperative Agreement No. 7200AA18CA00011. The personal protection equipment (PPE), such as face masks, was procured and provided by the Government of Guinea as part of the COVID-19 response. FS and AD are employees of Ministry of Health, Guinea. The funders had no role in the study design, data collection and analysis, decision to publish, or preparation of the manuscript.

pandemic. The epidemiological coverage was ≥65% for lymphatic filariasis, onchocerciasis and soil-transmitted helminths in 10 out of 19 HDs and ≥75% for schistosomiasis for school-aged children in 7 out of 11 HDs.

## Conclusions/significance

Guinea was one of the first countries in Africa to resume MDA activities during the COVID-19 pandemic without causing an observed increase of transmission. The development of a risk mitigation plan and a method to monitor adherence to barrier measures was critical to this unprecedented effort. The rapid incorporation of COVID-19 barrier measures and their acceptance by CDDs and household members demonstrated both the adaptability of the National NTD Program to respond to emerging issues and the commitment of the MoH to implement NTD programs.

## Author summary

Following the guidance from World Health Organization (WHO) on April 1, 2020 in the COVID-19 pandemic, the Guinea Ministry of Health (MoH) suspended mass drug administration (MDA) for neglected tropical diseases. WHO updated guidance on the resumption of field-based NTD activities on July 27, 2020 and the Guinea MoH and its partners planned and resumed MDA in mid-August to September 2020 in 19 health districts. The risks related to the process of MDA were assessed and a contingency plan and tools were developed to guide MDA implementation during the pandemic, adopting risk mitigation measures, such as testing for all national staff leaving Conakry, mask wearing, social distancing of 2 meters, and hand washing. A supervision checklist was used by supervisors to monitor compliance with these measures by the community drug distributors (CDDs) and household members during door-to-door MDA. It was shown that 68.3% of CDD teams practiced hand washing or sanitizing, compared to 45.0% by household members, and 64.2% of CDD teams respected social distancing of two meters during their distribution at households. There were 79.2% of CDD teams and 23.3% of household members wearing face masks. Among household participants 87.5% did not touch the dose pole and among CDD teams 88.3% did not touch the hands of the participants while giving the drugs. CDD teams and household participants (both 94.2%) were equally willing to participate in the MDA despite the COVID-19 context. The treatment coverage for lymphatic filariasis, onchocerciasis and soil-transmitted helminths was more than 65% in 10 out of 19 HDs and that for schistosomiasis was more than 75% for SAC in 7 out of 11 HDs. Guinea was one of the first countries in Africa to successfully resume MDA activities during the COVID-19 pandemic. The development of a risk mitigation plan and a method to monitor adherence to barrier measures was critical to this unprecedented effort.

## Background

Neglected tropical diseases (NTDs) are a diverse group of infectious diseases predominately found in tropical and subtropical areas. They affect more than one billion people worldwide, disproportionately those living in poverty, and account for approximately 26 million

disability-adjusted life years (DALYs) [1,2]. In addition to their impact on health, NTDs contribute to an immense social and economic burden resulting from social stigma, physical disabilities, disfigurement, blindness, discrimination, loss of social status, malnutrition, growth failure, and impaired cognitive development [2].

The World Health Organization (WHO) promotes several strategies to control and eliminate NTDs, including preventive chemotherapy [3]. The most prominent NTDs amenable to preventive chemotherapy (PC) include lymphatic filariasis (LF), onchocerciasis (OV), schistosomiasis (SCH), soil-transmitted helminthiasis (STH) and trachoma. The main approach for the control and/or elimination of this group of NTDs is the periodic administration of efficacious, safe, and inexpensive (usually donated) drugs to entire at-risk populations [4]. This approach, known as mass drug administration (MDA), must be implemented continuously (usually once per year) and must achieve sufficient population coverage (effective coverage) for a number of years until disease-specific targets are met [4]. Interruption of annual MDAs may prolong the process of achieving the control and/or elimination targets [5].

LF, OV, SCH, STH and trachoma are geographically widespread in Guinea. At least one of these NTDs is endemic in each of the 38 country's health districts (HDs), putting over 7 million people at high risk of contracting one or more of these diseases [6,7]. The overall objectives of Guinea's National NTD Program (NTDP) are to eliminate LF and trachoma as public health problems by 2030 and 2024, respectively; to eliminate transmission of OV by 2025; and to control morbidity caused by SCH and STH by 2025 [7]. In alignment with WHO recommended strategies [8], Guinea conducts annual MDA with azithromycin for trachoma, ivermectin for OV, combined ivermectin and albendazole for LF (and OV and STH), praziquantel for SCH and albendazole for STH. The objectives of MDA are to achieve effective program and epidemiological coverage and 100% geographic coverage yearly.

Steady progress has been made in delivering interventions to control and eliminate NTDs in Guinea. By 2016, all PC NTDs had been mapped and MDA for each disease had been scaled-up to full geographic coverage, where needed. To date, 15 of 18 trachoma-endemic HDs have achieved the criteria to stop MDA [9]. Additionally, all 24 LF-endemic HDs have implemented at least four rounds of MDA and are ready for or close to conducting epidemiological assessments to determine if LF elimination targets have been achieved. OV, STH, and SCH-endemic HDs continue to implement MDA in HDs endemic for each disease.

Along with these successes, Guinea has experienced significant challenges in implementing continuous rounds of MDA. For example, from 2013–2015, some rounds of MDA were not conducted due to national elections, political instability, and the Ebola virus epidemic. In 2019, all MDA activities were suspended due to social unrest caused by adverse events following the distribution of praziquantel in three HDs (Coyah, Dubréka and Fria). Most recently the COVID-19 pandemic has posed a major constraint for the safe and timely implementation of NTD interventions.

The NTDP began preparing for its 2020 MDA campaigns prior to the report of the first COVID-19 case in Guinea. By the time the first case was reported on March 12, 2020, the annual training-of-trainers had been completed in most HDs and NTD drugs had been delivered from the central level to each of the regions. Soon thereafter, a national state of emergency was declared, all land borders were closed, curfews were imposed, schools were closed, public gatherings were limited to 20 people, and mask-wearing became compulsory. Many health activities were paused [10]. On April 1, 2020, in an effort to reduce the risk of COVID-19 transmission associated with large-scale community-based health interventions, the WHO recommended that mass treatment campaigns, active case-finding activities and population-based surveys for NTDs be postponed until further notice [11]. Following this guidance, the

Act to End Neglected Tropical Diseases | West Program (Act | West) and the Guinea MoH suspended all fieldwork required to prepare for MDAs.

To plan for the resumption of NTD interventions in the context of COVID-19, and preserve the progress made to date in Guinea, the NTDP and Act | West developed a national contingency plan. This plan focused on 1) a risk-benefit assessment to identify potential risks and actions to mitigate them, 2) a risk mitigation plan to guide implementation of barrier measures during MDA, and 3) a remote supervision plan to monitor compliance with COVID-19 barrier measures in real-time during MDA.

Following further guidance from the WHO on the restart of NTD field activities [11], the Guinea MoH and Act | West resumed and completed MDA activities with strict risk mitigation measures in August–September 2020. A supervision checklist was created and used in the field supervision. The objectives of the study were to assess how the MDA was conducted during the COVID-19 pandemic with preventive and barrier measures and to assess the compliance to MDA and the COVID-19 mitigation measures during MDA. Here we present the experiences and lessons learned from preparing for and implementing MDA in the context of the COVID-19 pandemic in Guinea.

## Methods

### Ethics statement

Guinea MoH implements the national program to control and eliminate NTDs according to the WHO guidelines. Supervision is part of the routine program activities of the national NTD program for program quality as a standard public health measure, which does not require ethical approval. During the COVID-19 pandemic, Guinea introduced the COVID-19 mitigation measures as public health emergency measures. The supervisor's checklist was used as a supervision tool during the routine program supervision to monitor the compliance of CDDs and participants with the national NTD program COVID-19 mitigation plan for MDA implementation. The use of a checklist was also considered a standard public health measure during the COVID-19 emergency and did not require specific ethical approval. Data were collected mostly by observation and some simple questions. Supervisors explained to participants the purpose of the supervision and the importance of observing the COVID-19 mitigation measures. Verbal consent was received from participants for giving their answers, due to complying with social distancing and avoiding contact. The CDDs or participants were free to decline to give answers. No personal information from observees or respondents was recorded.

### Risk assessment

The National NTD Program, with partners' support, initiated and completed a COVID-19 transmission risk assessment to identify potential risks associated with the planned MDA during the COVID-19 pandemic in order to help stakeholders to avoid further transmission of the virus. The risk assessment considered the epidemic situation in each HD, including the reported COVID-19 cases, and the type of activities required to conduct MDA, such as CDD training and venue, drug distribution platforms (i.e. community-based MDA through door-to-door or fixed-point distribution, or school-based MDA), supervision, and travel of individuals from Conakry—the epicenter of COVID-19 in Guinea. The assessment also addressed risk factors leading to the community refusal to MDA in the pandemic context. Following the risk assessment, a mitigation plan was developed to enable MDA implementation for each step.

## COVID-19 mitigation plan

Following the risk assessment described above, a risk mitigation plan was then developed in accordance with national COVID-19 prevention guidance to reduce the risk of COVID-19 transmission during the MDA. All activities associated with the MDA implementation strategy, including social mobilization, training and drug distribution, were adapted. There was also mandatory COVID-19 testing for all participants leaving Conakry. Additionally, it was recommended that the vehicles travelling from Conakry have a maximum of two supervisors plus one driver in one vehicle.

## Social mobilization

Social mobilization materials on MDA and COVID-19 prevention were developed and provided to CDDs during their training. Messages were shared before and during the campaign in local languages at all levels through rural radio stations, and village town criers were involved in message dissemination. To mitigate the fear of population, additional messages relating to COVID-19 were added to those usually broadcast by town criers. For this, the town criers were briefed by the heads of health centers to reassure the population and invite them to take the NTD drugs. The key messages included social distancing of two meters, hand washing and wearing face masks. The routine national campaign launch ceremony was cancelled to prevent the gathering of large crowds. At the HD level, the launch ceremonies took place in the local chiefs' houses and health centers in compliance with the COVID-19 barrier measures, as recommended by the NTDP. Overall, all media were informed and trained on COVID-19 by the MoH well before the MDA campaign.

## Training

The training of trainers was completed in Guinea before the first case of COVID-19 in the country. Therefore, no specific plan was developed for the training of trainers. However, for the training of CDDs, the training was conducted in groups of <30 people and in an open space or in a classroom with sufficient aeration and ventilation with open windows. During the training, all participants had to comply with the requirements including social distancing of two meters and wearing a mask in the training room. Handwashing stations provided by the government and its partners were positioned in front of each training site. As large grouping was prohibited, all participants (trainers as well as CDDs) had mandatory requirements to avoid congregating in groups when going to the training sites. Refreshments were not provided to the group during the break and participants were provided an allowance instead.

## Mass drug administration

MDA was carried out by the NTDP and its partners with a significant focus on COVID-19 prevention measures. It took place in 19 HDs as initially planned with the support of Act | West Program. These included 11 HDs treated with the combination of ivermectin, praziquantel and albendazole (IPA) and 8 HDs with ivermectin and albendazole (IA).

During previous MDAs, Guinea used both fixed-point and door-to-door community-based strategies and school-based drug distribution approach. Due to the strict measures required in the COVID-19 context, the community-based drug distribution strategy only utilized the door-to-door approach to treat people. A school-based strategy also commonly utilized during MDA in Guinea was not possible since schools were closed.

All CDDs, supervisors and town criers involved in the MDA were provided with locally-made (fabric) face masks by the MoH. Soap was provided for handwashing by the NTDP.

CDDs were required to wash their hands before distributing drugs and were trained to inform household members to wash their hands before taking the drugs. Usually, CDDs directly measured the height of participants using the dose poles in their hands with close physical distance. However, in the COVID-19 context, the CDDs were instructed to attach dose poles to a wall, a tree or a standing object and ensured that the dose poles were not touched by participants during height measurement. CDDs were trained to measure the participants from a distance and to distribute the required number of tablets using the cap of the drug container or a spoon without touching the hands of participants.

## Supervision and checklist

A COVID-19 preventive and barrier measures checklist was developed to monitor compliance with COVID-19 barrier measures during MDA by field supervisors. The checklist included: 1) COVID-19 context related information, 2) CDDs and household participants' practices during drug administration, and 3) perception of community and the supervisors (see S1 Checklist). Central level program staff and supervisors were tested for COVID-19 infection before leaving Conakry and only those with negative test results were allowed to travel to the districts to conduct the supervision. During supervision, while observing CDDs on the quality of drug distribution, supervisors also observed CDDs on their compliance with COVID-19 mitigation measures according to the checklist. While using the COVID-19 checklist, the national supervisors were able to confirm in real time whether the CDDs were complying with the COVID-19 mitigation measures during MDA. They were also able to remind field supervisors to comply with the measures during supervision and to make corrective instructions to the CDDs as needed.

## Study population and sampling

Five of 19 HDs started the MDA before the checklist was available. The checklists to monitor compliance with preventive measures and barriers against COVID-19 were used in 14 of the 19 HDs. Thus, the results presented here relate to adherence to the contingency plan for the implementation of MDA in the context of the COVID-19 epidemic in 14 HDs. The study population included CDDs in the 14 HDs and household members they visited. National supervisors were instructed to complete one checklist for each of CDD teams (two CDDs per team) they supervised without a specific quota. Supervisors used convenience sampling to select CDD teams for supervision at their convenience in the field without informing the selected CDDs in advance. One checklist was completed for each CDD team observed conducting MDA at one household (a household is defined as a house compound with one or several persons who live together and share meals). In total, 120 checklists for 120 different CDD teams (representing 240 CDDs) at 120 households were completed.

## Data collection and analysis

CDDs conducted a population census in communities during the MDA and used registers to record treatments for their catchment communities. MDA data were collected with these registers and summarized treatment numbers were reported to the NTDP and transferred to an S1 Database. The MDA supervision information (registers filling, treatment record, respect of eligibility, dose pole usage etc.) was collected on paper as usual, while the compliance of COVID-19 preventive and barrier measures checklist was completed through electronic data capture using the open data kit (ODK)-based ONA platform.

Data were analyzed using Excel and Epi Info 7. Descriptive analysis was done for district MDA numbers and coverage. The epidemiological coverage of MDA for LF, OV and STH was

calculated using projected total population in each district according to the national census 2017 [12,13] as denominators, while treatment coverage for SCH was calculated using the projected numbers of school-age children (SAC) from the population as denominators. The variables of interest were COVID-19 epidemic situation in the district, the practice of preventive and barrier measures during MDA observed by supervisors (mask wearing, hand washing, dose pole using, social distancing, etc.), and the reluctance of communities and CDDs to participate in MDA. We used an uncorrected Chi-squared test or Fisher's exact test to compare adherence to COVID-19 mitigation measures between HDs with and without confirmed COVID-19 cases.

## Results

### MDA coverage

Table 1 summarizes the MDA results in 19 HDs. The epidemiological coverage rate for mass treatment for LF, OV and STH varied from 53% to 79%. Using the WHO-recommended required effective coverage ($\geq$65%) threshold for LF [13], the epidemiological coverage rate was $\geq$65% in 10 out of 19 HDs. Among the 9 districts where the coverage was <65%, 8 had reported positive cases of COVID-19 before the campaign. Among the 10 districts with a coverage $\geq$65%, 7 health districts had also reported positive cases of COVID-19. In total 4,548,942 people were treated for LF, OV and/or STH, among whom 53% were women (2,412,686/

**Table 1. MDA coverage results in 19 health districts in 2020 in Guinea.**

| District | Total population | No. of persons treated for LF/OV/STH | Epidemiologic coverage for LF/OV/STH | No. of persons treated for SCH | Program coverage for SCH |
|---|---|---|---|---|---|
| Boke* | 537,812 | 399,818[a,b] | 74.3% | N.D. [c] | - |
| Gaoual | 231,483 | 139,723 [b] | 60.4% | 48,288 | 74.5% |
| Koundara | 155,302 | 104,046[b] | 67.0% | N.D. [c] | - |
| Dabola | 216,594 | 153,127[b] | 70.7% | N.D. [c] | - |
| Dinguiraye* | 234,849 | 165,282[b] | 70.4% | 82,459 | 125.4% |
| Faranah* | 334,996 | 194,227 | 58.0% | 74,392 | 79.3% |
| Kissidougou* | 338,718 | 189,311 | 55.9% | 66,949 | 70.6% |
| Kankan* | 563,884 | 365,921 | 64.9% | N.D. [c] | - |
| Kerouane* | 248,174 | 188,207 | 75.8% | 47,472 | 68.3% |
| Kouroussa* | 320,952 | 169,128 | 52.7% | N.D. [c] | - |
| Mandiana* | 401,719 | 257,104 | 64.0% | N.D. [c] | - |
| Siguiri* | 810,208 | 523,784 [b] | 64.6% | 171,772 | 75.7% |
| Forecariah* | 290,227 | 194,286 | 66.9% | N.D. [c] | - |
| Kindia* | 524,886 | 416,391 | 79.3% | 150,563 | 102.4% |
| Telimele* | 339,969 | 255,558[a] | 75.2% | N.D. [c] | - |
| Lelouma* | 194,774 | 106,337 | 54.6% | 38,949 | 71.4% |
| Beyla | 389,932 | 287,678 | 73.8% | 115,046 | 105.4% |
| Gueckedou* | 347,284 | 229,384 | 66.1% | 74,961 | 77.1% |
| Macenta* | 332,946 | 209,630 | 63.0% | 81,289 | 87.2% |

Notes

* Districts with reported COVID-19 cases at the time of MDA

[a] Districts not treated for OV

[b] Districts not treated for STH

[c] N.D. districts not treated for SCH.

4,548,942), and 35.8% were school-aged children (SAC) (1,627,618 /4,548,942). In 11 districts targeted for SCH, the reported coverage ranged from 68.3% to 125.4%. Seven districts achieved the minimum required 75% as recommended by the WHO, while 4 districts fell below 75%.

### Adherence to COVID-19 preventive and barrier measures

There were confirmed positive COVID-19 cases in 15 of the 19 MDA health districts. The number of positive cases detected in the districts ranged from 1 to 504 at the time of the campaign.

Supervisors observed CDD teams distributing drugs at households and information on adhering to the COVID-19 mitigation measures according to the supervision checklist was collected and reported. A total of 120 checklists were completed, representing 120 CDD teams at 120 households. Washing or disinfecting hands when entering and leaving house compounds was practiced by CDD teams in 82 of 120 occasions (68.3%), compared to 54 of 120 occasions (45%) among households during the MDA campaign (Fig 1). Face masks to cover the mouth and nose were worn by 79.2% (95/120) of CDD teams (Fig 1), compared to a lower 23.3% among households (28/120) (Fig 1). The social distancing of two meters was respected by 64.2% (77/120) of CDD teams during their distribution at households.

In 105 (87.5%) out of 120 observations, household participants did not touch the dose pole while taking their height measurement to estimate the dose for medication. In 106 observations out of 120 (88.3%), CDDs did not touch the hand of the participants while giving the drugs.

A large majority of CDD teams (94.2% [113/120]) and household participants (94.2% [113/120]) were willing to participate in the MDA campaign despite ongoing COVID-19 pandemic.

Among the 120 observations, 20 were from districts without confirmed COVID-19 cases while 100 were from districts with confirmed COVID-19 cases. Adherence to the mitigation measures in districts where COVID-19 cases were reported was similar to that in districts

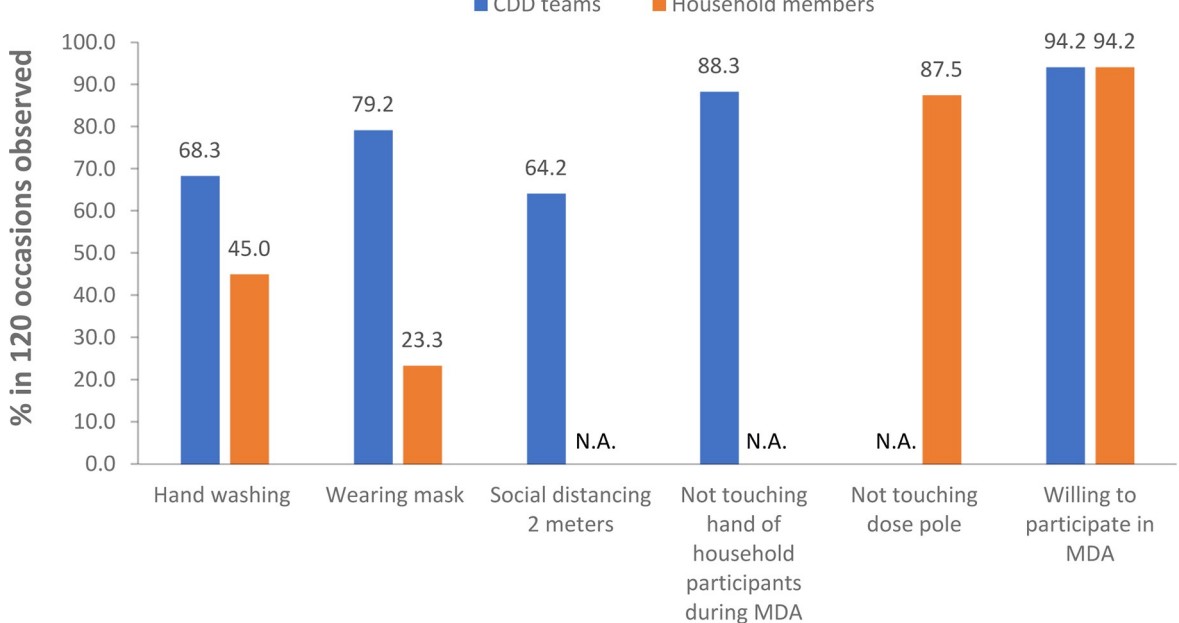

**Fig 1. Adherence to COVID-19 preventive and barrier measures by the CDD teams and household participants during MDA in 14 health districts in 2020 in Guinea.**

**Table 2. Observative results on adherence to COVID-19 mitigation measures during MDA from 14 districts in 2020 in Guinea.**

| Adherence to COVID-19 preventive and barrier measure during MDA | District without COVID-19 confirmed cases | District with COVID-19 confirmed cases | P value |
|---|---|---|---|
| Hand washing by CDDs team | 70% (14/20) | 68% (68/100) | 0.86 |
| Hand washing by household participants | 45% (9/20) | 45% (45/100) | 1 |
| Wearing mask by CDDs | 95% (19/20) | 76% (76/100) | 0.07* |
| Wearing mask by household participants | 10% (2/20) | 26% (26/100) | 0.12 |
| Physical distancing two meters | 60% (12/20) | 65% (65/100) | 0.67 |
| Touching dose pole by household participants | 0% (0/20) | 15% (15/100) | 0.06 |
| Touching hand of household participants during MDA by CDDs | 15% (3/20) | 11% (11/100) | 0.61 |
| CDDs reluctant to participate in MDA | 0% (0/20) | 7% (7/100) | 0.22 |
| Household participants reluctant to participate in MDA | 15% (3/20) | 4% (4/100) | 0.06 |

Note

* Fisher's exact test.

where no case of COVID-19 was reported (Table 2). We did not find any statistically significant difference in practices or willingness to participate in MDA between CDDs and household members according to district status (with or without cases of COVID-19).

## Discussion

Guinea was one of the first countries in Africa to resume routine MDA activities during the COVID-19 pandemic. The successful restart of MDA implementation in the context of COVID-19 was a joint effort of the NTDP and its partners. The development of a risk mitigation plan and a method to monitor adherence to barrier measures were critical to this unprecedented effort.

The results demonstrate that implementation of MDA in the context of COVID-19 is programmatically challenging, but with appropriate planning, resources, and training, risk mitigation measures can be rapidly applied. Central to this effort was the commitment of the MoH to 1) procure personal protection equipment (PPE) for program staff and CDDs involved in the MDA, and 2) put in place a COVID-19 test rule requiring all fieldworkers and supervisors leaving the capital city of Conakry (the national COVID-19 epicenter) to be tested prior to travel. To the authors' knowledge, this was a unique requirement not replicated elsewhere for MDA. The participatory approach, led by the NTDP, to develop the contingency plan contributed to wide acceptance of restarting the MDA from all key stakeholders (such as administrative, political and religious leaders at different levels in country, health personnel, donors and implementing partners, etc.).

MDA coverage was below what would be expected under non-pandemic circumstances, but given the context, we view this MDA as a success. Ten out of 19 HDs reached the minimum required coverage recommended by the WHO for LF (≥65% epidemiological coverage) and 7/11 HDs for SCH targeting SAC (≥75%) based on national census data. As the treatment was integrated for LF, OV and STH in some districts (Table 1), we used the coverage for LF as a representative for both LF, OV and STH. Separate coverage or treatment numbers for SAC for STH in such integrated MDA was not reported.

The vast majority (94.2%) of CDDs and household members observed were willing to participate in the MDA despite COVID-19. Poor coverage in certain districts may be explained by multiple reasons. Firstly, eight out of nine HDs of LF districts which did not achieve the minimum coverage had reported cases of COVID-19. Districts with reported cases of COVID-19

were similar in respecting the COVID-19 preventive and barrier measures during the MDA to districts where no case was confirmed (Table 2) and there was no difference in willingness between households and CDDs to participate in the MDA regardless the status of the districts. However, there was indeed a small proportion (5.8%, 7/120) of households with members who were reluctant to participate in the MDA, though this figure was confined to accessible areas (to supervisors) and may not reflect the reluctance in the overall population. Secondly, it is important to note that the MDA happened during the rainy season. It was originally planned in March and was delayed due to COVID-19 suspension but had to be implemented in August and September to avoid drug expiration. During the rainy season, access to the target population in some places is difficult and people are not often in the households during the MDA campaign.

Overall, the preventive and barrier measures were followed by CDD teams more than household members (Fig 1). Washing hands was followed by 82 of the 120 (68.3%) CDD teams against 54 of the 120 (45.0%) households and wearing masks was followed by 95 of the 120 (79.2%) CDD teams against 28 of the 120 (23.3%) households. CDDs had received masks from the MoH and it was mandatory for the population to wear a mask according to the government policy to fight COVID-19 [10]. In 15 of the 120 (12.5%) households supervised, some household participants were found to have touched the dose pole while they were having their height measured and in 14 of the 120 (11.7%) CDD teams supervised, CDDs were found to have touched the hand of household participants while distributing drugs. That a significant number of CDD teams and household members did not follow the rules for mitigation measures highlighted the challenge and need for more intensive training and communication on COVID-19 in changing perception and behavior in CDDs and communities. Infrequent use of face masks by household members may also highlight the lack of availability of face masks for households in rural communities. The MDA strategy (door-to-door) helped crowd control, and social distancing of two meters was observed in 64.2% of observations. Supervisors provided corrective suggestions when noncompliance with COVID-19 mitigation measures was observed during the supervision.

All the components and materials of the MDA campaign were reviewed as indicated in the methodology. The training was conducted in small groups for CDDs, and the COVID-19 module was added to the initial training agenda. These increased the workload of trainers in terms of extra number of sessions and duration of training. In fact, for the health centers with a high number of CDD teams (more than 30), the number of training sessions increased although they were held during the two days planned for the training. Thus, the heads of the concerned health centers and their deputies organized two training sessions per day with a two-hour break throughout the two days, compared with one training session on one day normally.

The increased measures to mitigate the risk of COVID-19 transmission during the drug distribution increased the workload of CDDs, with the adoption of the door-to-door strategy as the sole method of distribution across all districts and for the first time CCDs were doing the census of the people during MDA. In addition, the CDD teams needed to ensure that the dose pole was not touched by the participants during measurement and that each household member washed or disinfected his or her hand before taking the drugs. This extended the time taken by each CDD team to complete a household. Normally from the NTDP data, the CDD team target is 150 people per day for an MDA in Guinea; however, with additional mitigation measures, one CDD team was only able to cover 90–110 people per day.

It is important to note that with appropriate mitigation measures, implementation of MDA during the COVID-19 pandemic did not cause an observed increase in COVID-19 transmission in Guinea. According to the published records, during the five weeks between 10 August

and 13 September 2020, the weekly total number of new cases reported were 690, 456, 333, 407 and 245 throughout Guinea and there was no increasing trend in new case reporting after 4 September when MDA ended [14]. We did not specifically follow CDDs on COVID-19 infection after MDA, but any COVID-19 infection among them would have been reflected in the national new cases reported during the period. Furthermore, none of the supervisors after MDA reported having symptoms of COVID-19 or being tested positive for COVID-19 infection.

There are certain limitations in the data presented here. Firstly, the sample size was relatively small. The supervision checklist was introduced a week into implementation to monitor COVID-19 barrier and preventive measures during the MDA and the adherence was not monitored during the early days of the MDA implementation. Some districts started the campaign before the tool was available. The supervision checklist was used in 14 of the 19 HDs conducting MDA and supervisors were only able to supervise a limited number of CDD teams with convenience samples. Therefore, the data may not have reflected the true situation across all MDA districts. Secondly, one checklist form completed represented one CDD team at one household. The results did not consider the number of people involved during the MDA at the household and the data at individual level. If one person did not observe the rules this was noted on the form and therefore the results may have underestimated the true compliance. And finally, data for households were collected at the households that were covered by CDDs and reasonably feasible for supervisors to visit; the households were not randomly selected (convenience sampling). Therefore, in this case, the data may have overestimated the community willingness to participate in MDA. The unwillingness to participate in MDA by some households may have contributed to the relatively poor coverage reported in some districts in addition to other factors mentioned above.

In conclusion, the rapid development of COVID-19 contingency plans for MDA demonstrates the adaptability of the Guinea NTDP and its partners to respond to emerging issues. It also reinforces the commitment of the MoH to implement NTD programs and–even in the face of a global pandemic. The Guinea NTDP was one of the first to successfully implement MDA in the context of COVID-19, with relatively good coverage. A high proportion of household members and CDDs (94.2%) interviewed were willing to participate in MDA. The COVID-19 preventive and barrier measures were generally followed on the ground by both CDDs and household members.

## Supporting information

**S1 Checklist. COVID-19 Barrier Measures during MDA.**
(DOCX)

**S1 Database. Adherence to COVID-19 mitigation measures during MDA.**
(XLSX)

## Acknowledgments

The authors thank the populations in the endemic districts, community drug distributors and supervisors for their participation in the mass drug administration. Thanks are also given to some members of the Act | West team at Helen Keller International and Family Health International 360 for their participation in discussions in the resumption of the Guinean NTD field activities.

The contents are the sole responsibility of authors and do not necessarily reflect the views of USAID, the United States Government or the Government of Guinea.

## Author Contributions

**Conceptualization:** Fatoumata Sakho, Christelly Flore Badila, Benoit Dembele, Aissatou Diaby, Abdoul Karim Camara, Lamine Lamah, Angel Weng, Yaobi Zhang, Angela Weaver.

**Data curation:** Christelly Flore Badila, Benoit Dembele, Abdoul Karim Camara, Lamine Lamah, Angel Weng.

**Formal analysis:** Benoit Dembele, Abdoul Karim Camara, Lamine Lamah, Angel Weng, Yaobi Zhang.

**Funding acquisition:** Fatoumata Sakho, Christelly Flore Badila, Aissatou Diaby, Steven D. Reid, Angela Weaver.

**Investigation:** Fatoumata Sakho, Christelly Flore Badila, Benoit Dembele, Aissatou Diaby, Abdoul Karim Camara, Lamine Lamah.

**Methodology:** Fatoumata Sakho, Christelly Flore Badila, Benoit Dembele, Aissatou Diaby, Abdoul Karim Camara, Lamine Lamah, Steven D. Reid, Angel Weng, Brian B. Fuller, Katherine A. Sanchez, Achille Kabore, Yaobi Zhang, Angela Weaver.

**Project administration:** Fatoumata Sakho, Christelly Flore Badila, Aissatou Diaby, Lamine Lamah, Steven D. Reid, Katherine A. Sanchez, Achille Kabore, Angela Weaver.

**Resources:** Katherine A. Sanchez, Achille Kabore, Angela Weaver.

**Software:** Angel Weng, Brian B. Fuller.

**Supervision:** Fatoumata Sakho, Christelly Flore Badila, Aissatou Diaby, Abdoul Karim Camara, Lamine Lamah.

**Validation:** Christelly Flore Badila, Benoit Dembele, Abdoul Karim Camara, Lamine Lamah, Angel Weng, Yaobi Zhang, Angela Weaver.

**Visualization:** Benoit Dembele, Yaobi Zhang.

**Writing – original draft:** Christelly Flore Badila, Benoit Dembele, Yaobi Zhang.

**Writing – review & editing:** Fatoumata Sakho, Christelly Flore Badila, Benoit Dembele, Aissatou Diaby, Lamine Lamah, Steven D. Reid, Angel Weng, Brian B. Fuller, Achille Kabore, Yaobi Zhang, Angela Weaver.

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
