## [Decision Letter · Decision Letter 0]

18 Jun 2021

Dear Dr DEMBELE,

Thank you very much for submitting your manuscript "Implementation of mass drug administration for neglected tropical diseases in Guinea during the COVID-19 pandemic" for consideration at PLOS Neglected Tropical Diseases. As with all papers reviewed by the journal, your manuscript was reviewed by members of the editorial board and by several independent reviewers. The reviewers appreciated the attention to an important topic. Based on the reviews, we are likely to accept this manuscript for publication, providing that you modify the manuscript according to the review recommendations. 

Sincerely,

Matthew C Freeman, MPH, Ph.D.

Associate Editor

Christine Petersen

Deputy Editor

Reviewer's Responses to Questions

**Key Review Criteria Required for Acceptance?**

**Methods**

-Are the objectives of the study clearly articulated with a clear testable hypothesis stated?

-Is the study design appropriate to address the stated objectives?

-Is the population clearly described and appropriate for the hypothesis being tested?

-Is the sample size sufficient to ensure adequate power to address the hypothesis being tested?

-Were correct statistical analysis used to support conclusions?

-Are there concerns about ethical or regulatory requirements being met?

Reviewer #1: - Are the objectives of the study clearly articulated with a clear testable hypothesis stated? yes but no hypothesis du to study design(report)

-Is the study design appropriate to address the stated objectives? Yes

-Is the population clearly described and appropriate for the hypothesis being tested? Yes only population is clearly described

-Is the sample size sufficient to ensure adequate power to address the hypothesis being tested? The sample signe like too small to be representative.

-Were correct statistical analysis used to support conclusions? yes

-Are there concerns about ethical or regulatory requirements being met? yes

Reviewer #2: Yes indeed the methods are well explained.

The authors need to better explain the following in the section of risk assessment “drug distribution platforms”

Please the following need more precision “travelers from Conakry avoid crowded vehicles.”

Please the following is a repetition and not well written “The MDA and prevention and protection against COVID-19 messages were disseminated by local media before and during the campaign.

Please rewrite the following part in social mobilization section “Some routine activities, such as the national campaign launch ceremony were cancelled to prevent the gathering of crowds.”

“Central level program staff and supervisors were tested for COVID-19 before leaving Conakry and only staff with negative results travelled to the districts and conducted the supervision. All supervisors observed COVID-19 mitigation measures during supervision visits, including wearing face masks, social distancing of two meters, and using hand sanitizer.” As a limit the authors did not indicated whether those negative while in the field respected confinement period before working because the newly infected will not be positive directly 

I will suggest to the authors to add study population section and sample size for more clarification

Reviewer #3: The objectives should be clearly articulated, preferably by stating the general objective and 2-3 specific objectives. These should be closely aligned to Similarly clear study questions. The discussion should then be restructured to speak to the objectives.

Reviewer #4: Minor revisions

- In the 1st sentence of the Methods section, the word “in” seems to be a typo, please double check it

- The break times are the riskiest periods during the training sessions, was there specific directions for these periods? If yes, please specify them

**Results**

-Does the analysis presented match the analysis plan?

-Are the results clearly and completely presented?

-Are the figures (Tables, Images) of sufficient quality for clarity?

Reviewer #1: -Does the analysis presented match the analysis plan? Yes

-Are the results clearly and completely presented? Yes but should be more clear to compare MDA coverage with the results of previous MDA to measure the impact of COVID-19.

-Are the figures (Tables, Images) of sufficient quality for clarity? Yes bu need to be review with comment above

Reviewer #2: Yes the results are well presented

“Thus, the results presented here relate to adherence to the contingency plan for the implementation of MDA in the context of the COVID-19 epidemic (barrier and preventive measures against COVID-19) in 14 HDs.” Read this part carefully something is missing”

Tables and figures are good

Reviewer #3: No revision required

Reviewer #4: Minor revisions

- In table 1, please explain the coverage > 100% or discuss them in the appropriate section

- Please check this sentence of the last paragraph of the Results section « Adherence to the mitigation measures for districts reporting COVID-19 cases was similar compared to districts where no case of COVID-19 was confirmed (Table 2). »

- The last sentence of the Results sections contains a word that need to be checked and replaced; the same sentence seems to consider a p value =0.05 as statistically significant. A p value should be <0.05 to be considered statistically significant.

- Some of the small numbers in the last table seems to require a Fisher Exact test instead of an uncorrected Chi square test (i.e. Household participants reluctant to participate in MDA line), Please double check

**Conclusions**

-Are the conclusions supported by the data presented?

-Are the limitations of analysis clearly described?

-Do the authors discuss how these data can be helpful to advance our understanding of the topic under study?

-Is public health relevance addressed?

Reviewer #1: -Are the conclusions supported by the data presented? Yes

-Are the limitations of analysis clearly described? Yes

-Do the authors discuss how these data can be helpful to advance our understanding of the topic under study? Yes but not sufficient

-Is public health relevance addressed? The study focuses on the proportions of actors who apply the barrier measures. It doesn't give much scientific interest. It would have been good to push the study by measuring the impact of the campaign on the occurrence of new cases of COVID-19 and especially if actors have developed signs after this campaign.

Reviewer #2: “The results demonstrate that implementation of MDA in the context of COVID-19 is programmatically challenging, but that with appropriate planning, resources, and training, risk mitigation measures can be rapidly applied.” I will suggest to remove “that”

Reviewer #3: No revision required

Reviewer #4: (No Response)

**Editorial and Data Presentation Modifications?**

Reviewer #1: Minor Revision

Reviewer #2: (No Response)

Reviewer #3: No revision required

Reviewer #4: Minor modifications

Abstract: 

- A definition of the world CDDs team and Households would be helpful in understanding these concepts.

- At least 2 sentences started with numbers 

Background

- Please add a reference at the end of the first paragraph of the Background section

- Please split the last sentence of the 3rd paragraph into two sentences to gain in clarity

Discussion

- At the end of the second paragraph of the Discussion sections, it would be informative to name the key stakeholders evocated.

- Please check the 2nd sentence of the 4th paragraph “Firstly, six out of seven HDs of LF districts which did not achieve the minimum coverage had reported cases of COVID-19. » for clarity

- In the same paragraph and elsewhere in the manuscript, the authors seem to be using interchangeably « households » and « households’ members », please look at that carefully and adapt according to the meaning of the concept in each sentence throughout the manuscript.

References

- Many references used are not eligible for a peer reviewed paper 

- References that are not peer reviewed journal articles should have a web link and date of access

- The types of documents eligible for citation as a reference should be check on the Journal’s website

**Summary and General Comments**

Reviewer #1: The study is interesting but remains in the form of a report. The authors could have extended this study by taking into account the impact of the implementation of barrier measures on the transmission of COVID-19. If they were effective, the results could be used as evidence to be strongly recommended to other countries.

Reviewer #2: Please revise this section size “The participatory approach, led by the NTDP, to develop the contingency plan contributed to wide acceptance of restarting the MDA from all key stakeholders.”

Please read this again not sure if this statement is correct as it is “and it was not possible to calculate STH coverage for SAC only” . my question is why not even if it was integrated normally I will be possible to report

This need also to be taken into account in the mitigation plan like asking and get agreement with communities for the best tie during the day distribute the drugs “During the rainy season, access to the target population and some places are difficult and people are not often in the household during the MDA campaign.”

Reviewer #3: The study is relevant to the current realities of COVID-19 and provides useful insights on how programs can implement interventions to safeguard gains already made, while taking necessary precautions to minimize the risk of getting infected with the disease.

Reviewer #4: This paper describes a real and current issue that threatened interventions targeting NTDs. It provides with an attitude that is well described and could help in the future when similar situation is faced.

PLOS authors have the option to publish the peer review history of their article (what does this mean?). If published, this will include your full peer review and any attached files.

Reviewer #1: No

Reviewer #2: No

Reviewer #3: Yes: Dr. Sultani Hadley Matendechero

Reviewer #4: Yes: Yaya Ibrahim Coulibaly

Figure Files:

Data Requirements:

Reproducibility:

References

---

## [Decision Letter · Decision Letter 1]

10 Sep 2021

Dear Dr DEMBELE,

We are pleased to inform you that your manuscript 'Implementation of mass drug administration for neglected tropical diseases in Guinea during the COVID-19 pandemic' has been provisionally accepted for publication in PLOS Neglected Tropical Diseases.

Best regards,

Matthew C Freeman, MPH, Ph.D.

Associate Editor

Christine Petersen

Deputy Editor

Reviewer's Responses to Questions

**Key Review Criteria Required for Acceptance?**

**Methods**

-Are the objectives of the study clearly articulated with a clear testable hypothesis stated?

-Is the study design appropriate to address the stated objectives?

-Is the population clearly described and appropriate for the hypothesis being tested?

-Is the sample size sufficient to ensure adequate power to address the hypothesis being tested?

-Were correct statistical analysis used to support conclusions?

-Are there concerns about ethical or regulatory requirements being met?

Reviewer #1: Are the objectives of the study clearly articulated with a clear testable hypothesis stated? Yes

-Is the study design appropriate to address the stated objectives? Yes

-Is the population clearly described and appropriate for the hypothesis being tested? Yes

-Is the sample size sufficient to ensure adequate power to address the hypothesis being tested? Yes

-Were correct statistical analysis used to support conclusions? Yes

-Are there concerns about ethical or regulatory requirements being met? Yes

Reviewer #2: the authors address all my previous comment and suggestions related to methods

Reviewer #3: The methods are clearly illustrated. Objectives are stated and well articulated.

**Results**

-Does the analysis presented match the analysis plan?

-Are the results clearly and completely presented?

-Are the figures (Tables, Images) of sufficient quality for clarity?

Reviewer #1: Does the analysis presented match the analysis plan? Yes

-Are the results clearly and completely presented? Yes

-Are the figures (Tables, Images) of sufficient quality for clarity? Yes

Reviewer #2: the authors address all my previous comment and suggestions related to results

Reviewer #3: The result are well presented and analyses.

**Conclusions**

-Are the conclusions supported by the data presented?

-Are the limitations of analysis clearly described?

-Do the authors discuss how these data can be helpful to advance our understanding of the topic under study?

-Is public health relevance addressed?

Reviewer #1: Are the conclusions supported by the data presented? Yes

-Are the limitations of analysis clearly described? Yes

-Do the authors discuss how these data can be helpful to advance our understanding of the topic under study? Yes

-Is public health relevance addressed? Yes

Reviewer #2: my concern about the conclusion part was also addressed

Reviewer #3: The conclusion is well captured and relevance to current circumstances clearly demonstrated.

**Editorial and Data Presentation Modifications?**

Reviewer #1: Accept

Reviewer #2: fine to me

Reviewer #3: (No Response)

**Summary and General Comments**

Reviewer #1: NA

Reviewer #2: i think the paper is now well presented and clear and will be helpful for others settings /countries to learn from this paper and prepare their mitigate plan for NTDS activities implementation under COVID 19 and others emergency infectious epidemic or pandemic.

Reviewer #3: All areas of previous concern have been adequately addressed.

PLOS authors have the option to publish the peer review history of their article (what does this mean?). If published, this will include your full peer review and any attached files.

Reviewer #1: No

Reviewer #2: No

Reviewer #3: **Yes: **Dr. Sultani Hadley Matendechero

---

## [Editor Report · Acceptance letter]

23 Sep 2021

Dear Dr DEMBELE,

We are delighted to inform you that your manuscript, "Implementation of mass drug administration for neglected tropical diseases in Guinea during the COVID-19 pandemic," has been formally accepted for publication in PLOS Neglected Tropical Diseases.

Best regards,

Shaden Kamhawi

co-Editor-in-Chief

Paul Brindley

co-Editor-in-Chief
